# Have Hand Hygiene Practices in Two Tertiary Care Hospitals, Freetown, Sierra Leone, Improved in 2023 following Operational Research in 2021?

**DOI:** 10.3390/tropicalmed8090431

**Published:** 2023-08-31

**Authors:** Matilda Mattu Moiwo, Gladys Nanilla Kamara, Dauda Kamara, Ibrahim Franklyn Kamara, Stephen Sevalie, Zikan Koroma, Kadijatu Nabie Kamara, Matilda N. Kamara, Rugiatu Z. Kamara, Satta Sylvia Theresa Kumba Kpagoi, Samuel Alie Konteh, Senesie Margao, Bobson Derrick Fofanah, Fawzi Thomas, Joseph Sam Kanu, Hannock M. Tweya, Hemant Deepak Shewade, Anthony David Harries

**Affiliations:** 1Ministry of Defence, Republic of Sierra Leone Armed Forces, Joint Medical Unit, Freetown 00232, Sierra Leone; gladysnanillak14@gmail.com (G.N.K.); stevesyllo@gmail.com (S.S.); 2Department of Clinical Research, Sustainable Health Systems, 34 Military Hospital Research Center, Freetown 00232, Sierra Leone; 3Ministry of Health and Sanitation, Freetown 00232, Sierra Leone; daudakamara50@gmail.com (D.K.); zikankoroma@gmail.com (Z.K.); kamarakadijatunabie@gmail.com (K.N.K.); spllenz54321@gmail.com (S.S.T.K.K.); smargao3@gmail.com (S.M.); samjokanu@yahoo.com (J.S.K.); 4World Health Organization Country Office, Freetown 00232, Sierra Leone; ibrahimfkamara@outlook.com (I.F.K.); derrickfbob@gmail.com (B.D.F.); 5College of Medicine and Allied Health Sciences, University of Sierra Leone, Freetown 00232, Sierra Leone; kamaramatilda9198@gmail.com (M.N.K.); thomasfawzi5@gmail.com (F.T.); 6United States Centres for Diseases Control and Prevention, Freetown 00232, Sierra Leone; rugiatuzkamara@gmail.com; 7Ministry of Agriculture and Forestry, Freetown 00232, Sierra Leone; samuelakonteh@gmail.com; 8Pharmacy Board of Sierra Leone, National Pharmacovigilance Centre, Freetown 00232, Sierra Leone; 9International Training and Education Centre for Health, Lilongwe P.O. Box 30369, Malawi; hmwtwea@yahoo.co.uk; 10Division of Health Systems Research, ICMR-National Institute of Epidemiology (ICMR-NIE), Chennai 600077, India; hemantjipmer@gmail.com; 11International Union Against Tuberculosis and Lung Disease (The Union), 2 Rue Lantier, 75001 Paris, France; adharries@theunion.org; 12Department of Clinical Research, Faculty of Infectious and Tropical Diseases, London School of Hygiene and Tropical Medicine, Keppel Street, London WC1E 7HT, UK

**Keywords:** hand hygiene opportunities, alcohol-based hand rub, hand wash with soap and water, World Health Organization hand hygiene observation tool, SORT IT, operational research, antimicrobial resistance, infection prevention and control (IPC), infections acquired in hospital

## Abstract

In 2021, an operational research study in two tertiary hospitals in Freetown showed poor hand hygiene compliance. Recommended actions were taken to improve the situation. Between February–April 2023, a cross-sectional study was conducted in the same two hospitals using the World Health Organization hand hygiene tool to assess and compare hand hygiene compliance with that observed between June–August 2021. In Connaught hospital, overall hand hygiene compliance improved from 51% to 60% (*p* < 0.001), and this applied to both handwash actions with soap and water and alcohol-based hand rub. Significant improvements were found in all hospital departments and amongst all healthcare worker cadres. In 34 Military Hospital (34MH), overall hand hygiene compliance decreased from 40% to 32% (*p* < 0.001), with significant decreases observed in all departments and amongst nurses and nursing students. The improvements in Connaught Hospital were probably because of more hand hygiene reminders, better handwash infrastructure and more frequent supervision assessments, compared with 34MH where interventions were less well applied, possibly due to the extensive hospital reconstruction at the time. In conclusion, recommendations from operational research in 2021 contributed towards the improved distribution of hand hygiene reminders, better handwash infrastructure and frequent supervision assessments, which possibly led to improved hand hygiene compliance in one of the two hospitals. These actions need to be strengthened, scaled-up and guided by ongoing operational research to promote good hand hygiene practices elsewhere in the country.

## 1. Introduction

Healthcare-associated infections are a major public health problem worldwide and are associated with increased hospital stay, long-term disability, increased resistance to antimicrobial drugs, poor clinical outcomes and unnecessary deaths [1,2]. Healthcare workers’ hands are crucially important with respect to transmitting the microorganisms responsible for healthcare-associated infections, and in the last 20 years, evidence has accumulated to suggest that improved hand hygiene can reduce healthcare-associated infections [3,4]. Efforts worldwide to reduce the frequency and burden of these infections have, therefore, focused on improved hand hygiene, including the implementation of the World Health Organization (WHO)’s multimodal hand hygiene strategy [5,6]. 

Hand hygiene practices can be assessed using a WHO standard observation tool [7], which outlines five opportunities or “moments” for observing hand hygiene within the healthcare setting. These include observing hand hygiene procedures before contact with patients and after contact with patients. This tool is widely used in various African countries and has found that, on average, there are poor levels of hand hygiene practices in various healthcare facilities in the different countries [8,9,10,11,12]. 

Sierra Leone has developed and launched a national strategic plan to combat antimicrobial resistance (2018–2022), as well as the National Infection Prevention Control (IPC) Guidelines (2022–2026). These guidelines clearly outline IPC activities, including hand hygiene practices, that should be implemented in healthcare facilities throughout the country [13]. Between June and August 2021, through an operational research project, hand hygiene practices were assessed using the WHO standard observational tool in two tertiary-level hospitals (34 Military Hospital—34MH—and Connaught Hospital) in Freetown, the capital city [14]. The findings from this study showed that less than 50% of hand hygiene opportunities were associated with compliance to accepted hand hygiene practices, with 34MH having a worse performance than Connaught Hospital. 

Various recommendations to improve hand hygiene compliance were made at the time in the published paper [14] and in the plain language hand-out used to disseminate findings to various stakeholders [15]. The current study was, therefore, carried out to assess how the findings of the 2021 operational research were disseminated [14], whether recommendations were acted on and whether these translated into improvement in hand hygiene compliance.

The overall aim of this study was to document and assess the dissemination and impact of hand hygiene compliance interventions recommended at two tertiary hospitals (34MH and Connaught Hospital) in Freetown, comparing the period June–August, 2021 with the period February–April, 2023. The specific objectives were to: (1) describe the dissemination activities resulting from the 2021 operational research findings, recommendations made and actions taken to promote hand hygiene compliance; and (2) compare overall hand hygiene compliance between 2021 and 2023 in the two hospitals and in relation to the five opportunities for hand hygiene action, the six different hospital wards and four different cadres of healthcare worker.

## 2. Materials and Methods

### 2.1. Study Design

For objective 1, this was a descriptive study of the dissemination, recommendations and action taken as a result of the first operational research study in 2021. For objective 2, this was a cross-sectional study using the standardised WHO observation tool to measure hand hygiene compliance [7].

### 2.2. Setting

#### 2.2.1. General Setting

Sierra Leone is a country on the coast of West Africa with a tropical climate and an environment that ranges from savanna to rainforests. The country has a total population of 7.5 million [16]. There are five administrative regions, within which there are sixteen districts. Sierra Leone has a healthcare system consisting of primary, secondary and tertiary care facilities under the control of the Ministry of Health and Sanitation. Health services are mainly provided by the public sector, with some of the services offered by private sector providers, non-governmental and faith-based organisations and the military establishment.

#### 2.2.2. Site-Specific Setting

The sites included in the study were 34MH and Connaught Hospital, both tertiary care hospitals situated in Freetown. Specifically, 34MH has approximately 120 beds and serves military personnel and their dependents, as well as the civilian population. Every year, 34MH admits approximately 2000 patients to its beds [17]. Connaught Hospital has approximately 300 beds and caters largely for the civilian population. Every year, Connaught Hospital admits approximately 4900 patients to its beds [18]. These two hospitals have been challenged in recent years with poor handwash infrastructure and inadequate access to clean running water. Veronica buckets were introduced to improve the water problem, although keeping these buckets filled up on a regular basis has not been easy. 34MH has also been undergoing extensive reconstruction in the last one year, which has compromised hand washing facilities and infrastructure.

Each hospital has IPC focal persons trained by the National Infection Prevention and Control Unit. These focal persons identify IPC link personnel (who are mostly nurses) in each ward to support IPC practices. One of the important responsibilities of the IPC focal persons is to carry out quarterly audits of hand hygiene compliance among the different cadres of healthcare worker.

### 2.3. Study Population

The study population included healthcare workers at the two tertiary hospitals (34MH and Connaught), Freetown, Sierra Leone, who were observed for hand hygiene compliance between February and April 2023 with their findings compared with those of health workers who were observed for hand hygiene compliance between June and August 2021.

### 2.4. Study Procedures 

#### 2.4.1. Dissemination Activities, Recommendations and Interventions 

Dissemination tools that included a plain language handout [15], a short (3-min) and long (10-min) power point presentation and an elevator pitch were developed during a Structured Operational Research Training Initiative (SORT IT) course module in May 2022 [19], which was held two months after the publication of the paper. A stakeholder list was also developed that prioritised key policy and decision makers to whom dissemination of the key findings should be carried out. Between May 2022 and January 2023, information was collected on the dissemination meetings held and their dates, the frequency of use of the dissemination tools and the number and cadre of key personnel who attended these meetings. Recommendations and actions that were proposed in the published paper and plain language hand-out [14,15] were documented.

#### 2.4.2. Hand Hygiene Compliance

In each hospital, the IPC link personnel observed hand hygiene compliance in the designated hospital wards. The healthcare workers observed included doctors, nurses, nursing students and laboratory technicians as they went about performing routine patient care.

The same methodology as was used in the previous study for observing and recording hand hygiene compliance [14] was used in this current study. In brief, before the data collection started in February 2023, a refresher training on how to use the WHO hand hygiene observation tool [7] was conducted for the IPC link personnel.

During each session (around 10–15 min), the IPC link person quietly (without the healthcare worker being aware) recorded the compliance to hand hygiene opportunities that included use of ABHR or hand washing. This silent observation was to prevent or reduce bias such as the Hawthorne effect, where people change their behaviour because they know and are aware that they are being observed [20]. We targeted 300–400 sessions, approximately the same number as were conducted in 2021 [14].

### 2.5. Data Variables and Sources of Data

The data variables were aligned to the study objectives.

For Objective 1, these included: number and type of dissemination meetings, trainings conducted for hospital staff and numbers of IPC personnel and staff trained; in the six hospital wards, observed on a monthly basis between February and April 2023, numbers of hand hygiene reminders and job aids on the tables and walls and the handwash infrastructure in place at that time; and supervisions, monitoring visits and hospital assessments conducted from May 2022. These data were obtained from records, personal observations and routine monitoring visits.

For Objective 2 and 3, these included: hospital (34MH, Connaught); date of hand hygiene session; duration of session in minutes; use of handwash or ABHR; the opportunities for hand hygiene actions and hand hygiene actions in the different hospital wards and amongst different healthcare worker cadres. The sources of data were the completed observation forms using the WHO standard hand hygiene observation tool.

### 2.6. Analysis and Statistics

We single-entered the data into Excel and exported this to STATA (version 18, StataCorp. 2023. Stata Statistical Software: Release 18. College Station, TX: StataCorp LLC.3.1) for analysis. Dissemination activities, recommendations and actions taken were described using frequencies and proportions. For hand hygiene variables, each hand hygiene opportunity was the unit of analysis. Hand hygiene compliance was calculated as a percentage by dividing the number of handwash actions taken by the number of opportunities for hand washing. The hand hygiene practices in this study for both hospitals and for the five opportunities for hand hygiene actions, for the six different hospital wards and the four different cadres of healthcare worker were compared with those observed and reported in the same two hospitals during June–August 2021 [14]. Comparisons were made using the chi-square test. The results were considered significant at the 5% level (*p* < 0.05, two-tail).

## 3. Results

### 3.1. Dissemination, Trainings, Recommendations and Actions following the 2021 Study

Dissemination activities included a National SORT IT dissemination meeting in November 2022 and two hospital meetings with hospital staff each at 34MH and Connaught between May and August 2022.

There were no funds for the residential training of healthcare workers on hand hygiene compliance. However, there were “on the job” trainings: one full-day-training on hand hygiene compliance at Connaught Hospital on all the wards with 30 staff (general staff and IPC staff) and three half-day trainings on hand hygiene compliance at 34MH with a total of 100–150 staff trained.

Between April 2022 and May 2023, the National Infection Prevention and Control Unit carried out 11 supervisory and IPC assessments at Connaught Hospital and six supervisory and IPC assessments at 34MH, with assessment grades being consistently ≥ 85%.

The recommendations and actions taken to improve hand hygiene compliance are shown in Table 1.

Observations were made each month between February and April 2023, with the recommended actions and interventions being the same at each of these times. Connaught Hospital performed better than 34MH with more hand hygiene reminders/job aids on walls and tables and more infrastructure available for ABHR and hand washing. Overall, there were 8 handwash stations per 100 beds in 34MH compared with 17 handwash stations per 100 beds in Connaught. Within each hospital, the department of medicine performed best with the surgical department a close second.

### 3.2. Comparison of Hand Hygiene Actions between the Two Hospitals

There were 7206 opportunities for hand hygiene actions observed over 327 sessions in 2023 compared with 10,461 opportunities observed over 423 sessions in 2021. 

Hand hygiene compliance overall and for ABHR and handwash with soap and water in the two hospitals between 2023 and 2021 is shown in Figure 1.

In 34MH (Figure 1a), there was a significant decline in hand hygiene compliance overall (40% to 32%, *p* < 0.001), although this only applied to handwash (20% to 13%, *p* < 0.001) and did not apply to ABHR (20% to 19%, *p* = 0.24). In Connaught Hospital (Figure 1b), there was a significant improvement in hand hygiene compliance overall (51% to 60%, *p* < 0.001), and this applied to both ABHR (27% to 32%, *p* < 0.001) and hand wash (23% to 28%, *p* < 0.001).

### 3.3. Hand Hygiene Actions in Relation to Five Opportunities

Hand hygiene actions in relation to the five opportunities for a hand hygiene action in each hospital are shown in Table 2.

In both hospitals, and as observed in 2021, the lowest hand hygiene compliance overall was before touching a patient or before conducting a clean/aseptic procedure, while the best hand hygiene compliance overall was after body fluid exposure. The two hospitals, however, differed significantly in performance. Hand hygiene compliance was significantly worse for all five opportunities in 34MH. In Connaught Hospital, hand hygiene compliance was significantly better before an aseptic procedure, after bodily fluid exposure and after touching a patient. There was no significant difference before touching a patient, but a significant decrease after touching a patient’s surroundings compared with 2021.

### 3.4. Hand Hygiene Actions in Relation to Hospital Departments/Wards

Hand hygiene actions in relation to hospital departments/wards in each hospital are shown in Table 3.

In 34MH, hand hygiene compliance significantly decreased in all departments/wards between 2021 and 2023. In contrast, in Connaught Hospital, hand hygiene compliance significantly increased in all departments/wards except for the paediatric ward and accident and emergency in 2023 compared with 2021.

### 3.5. Hand Hygiene Actions in Relation to Type of Healthcare Worker

The hand hygiene actions in relation to type of health worker in each hospital are shown in Table 4. In 34MH, there was a mixed picture when comparing 2023 with 2021, with hand hygiene compliance significantly improving in doctors but significantly decreasing in nurses and nursing assistants. In contrast, in Connaught Hospital there was a significant improvement in hand hygiene compliance in all healthcare workers between 2021 and 2023. Doctors particularly showed a significant improvement in hand hygiene compliance between 2021 and 2023 in both hospitals. 

## 4. Discussion

The key finding of this operational research study in Freetown, Sierra Leone, was the striking difference in hand hygiene compliance between the two study periods and across the two tertiary hospitals. First, 34MH showed an overall decrease in hand hygiene compliance, mainly due to poor hand washing actions rather than poor use of ABHR, while Connaught Hospital showed an overall increase in compliance with both ABHR and hand washing. Why were there such differences between the two hospitals?

Dissemination activities, improved awareness and “on the job” trainings took place in each hospital. However, at the time of the study, 34MH was undergoing extensive reconstruction, and this might have been responsible for the generally poor placement of hand hygiene reminders and job aids on the walls and tables of the different hospital wards, the deficiencies in hand wash stations, the complete absence of running water or paper towels and interruptions of ABHR in the paediatric wards. In contrast, in Connaught Hospital, there was much better placement of hand hygiene reminders and job aids, a two-times greater distribution of hand wash stations per 100 hospital beds and running tap water available on all wards. Supervision and IPC assessments were carried out twice as frequently at Connaught Hospital compared with 34MH. Furthermore, IPC link personnel changed in 34MH during the two years, while in Connaught Hospital they mostly remained the same, thus, providing continuity of hand hygiene oversight. There were at least two IPC focal points at Connaught Hospital compared with one at 34MH, and there was a stronger hospital antimicrobial resistance (AMR) committee established at Connaught Hospital. 

There is good evidence that WHO’s five-point multimodal strategy of system and improved infrastructure, education and training of healthcare workers, evaluation and feedback, health facility hand hygiene reminders and an institutional climate of safety can lead to a large, quick and sustained improvement in hand hygiene compliance in hospital settings [21]. However, solely addressing better education, improved knowledge and more awareness is not enough on its own to change hand hygiene behaviour [22]. Studies in various African hospital settings have found a significant and positive association between improved hand hygiene practices and the presence of hand hygiene reminders/job aids/handwash stations/and the uninterrupted presence of running water and soap [8,23,24,25,26]. These findings support our assumptions that workplace reminders, supportive hand hygiene infrastructure and more frequent supervision possibly contributed to better practices at Connaught Hospital. On that note, paper towels are another well-established important contributor to hand hygiene [27]. These were not available in either of the Freetown hospitals, and if they had been, this might have led to better hand hygiene compliance in both.

There were a number of other findings from this comparative study. Similar to the situation in 2021, we found that hand hygiene compliance in both hospitals was much better after contact with patients (self-protected opportunities) compared with before contact with patients (patient protective opportunities). These findings are in line with previous reports from other African hospitals [10,11,28,29,30,31] and from two secondary hospitals in Sierra Leone [32], where healthcare workers more readily performed hand hygiene actions to self-protect rather than for patient protection. In busy wards with high patient workloads, hand hygiene compliance is reported to decrease significantly [33]. It may be that in these situations, once healthcare workers have washed hands after touching a patient, they may feel it is unnecessary to repeat the procedure before seeing or touching the next patient. This is an area that requires qualitative research to understand the behaviour.

Hand hygiene compliance deteriorated in all hospital departments and wards at 34MH, with compliance being uniformly between 30–33%, while at Connaught Hospital there was an overall improvement in all departments with compliance in medicine, surgery, paediatrics and intensive care being above 60%. The difference between the two hospitals may again reflect the far better distribution of hand wash stations and hand wash infrastructure in the Connaught Hospital wards compared with those in 34MH. While previous studies in African settings have shown little difference between departments and wards [28,30], a recent study in Eastern Ethiopia showed an improvement in hand hygiene actions in medical and surgical wards over time [34], findings that are in line with what we found at Connaught Hospital over the two-year period.

In both hospitals in Freetown, it was encouraging to see an improvement in hand hygiene compliance in doctors. Previous studies have found that medical doctors are less likely to practice good hand hygiene compared with other cadres of staff [35,36]. In African settings particularly, hand hygiene compliance in doctors has been consistently low and inferior to nurses and other healthcare worker cadres [10,28,30,37,38]. The reasons for this improvement in doctors in our study are unclear. They might be related to COVID-19 pandemic behavioural change or because of the involvement of medical doctors in the implementation of IPC programs in the hospitals. All of these points deserve further investigation, as is the need to understand why hand hygiene compliance in nurses in 34MH declined so significantly between the two years.

The strengths of this study were the documentation of dissemination activities and actions taken following the 2021 research study, the large number of hand hygiene opportunities observed, the use of the same WHO hand hygiene observation tool in the two studies and the conduct and reporting of the study according to Strengthening the Reporting of Observational Studies in Epidemiology (STROBE) [39]. 

However, there were a few limitations. Due to time constraints in 2023, we were not able to match the 10,000 hand hygiene opportunities that we observed in 2021, although we came close at just over 7000. As stated previously, hand hygiene compliance decreases with increasing patient workload [33], and it would have been helpful to have measured patient workload in the two hospitals and between the different hospital wards. Despite taking precautions in silent observation on hand hygiene practice, we cannot completely exclude the Hawthorne effect [40]. Finally, we focused these two studies in 2021 and 2023 on two tertiary hospitals and our findings may not be generalizable to other facilities in Sierra Leone.

In spite of these limitations, there are three important implications and recommendations. First, although we are not in a position to attribute cause and effect, it would appear from our findings in Connaught Hospital that hand hygiene reminders, job aids and good handwash infrastructure contributed towards improved hand hygiene compliance on the ground. Sierra Leone is also a local producer of ABHR, placing the country in a very good position to take this important intervention forward and make it widely available throughout the wards in all hospitals in the country. External supervision and hospital assessments need to particularly focus on whether there is adequate handwash infrastructure.

Second, despite Connaught Hospital having apparently sufficient handwash infrastructure, healthcare workers failed to practice hand hygiene actions in 40% of opportunities, so improvements are needed. We previously suggested new ways to improve hand hygiene actions such as use of “emojis” or “positive nudges”, interdepartmental competition, positive reinforcements and so on [14]. In Nigeria, the installation of voice reminders in hospital wards significantly improved hand hygiene compliance [41], and in Finland, regular observations and feedback over a seven-year period in medical and surgical wards resulted in improved compliance amongst doctors and nurses [42]. These innovations could be added to the list.

Third, while trying to improve hand hygiene practices in the two tertiary hospitals in the study, more needs to be carried out to expand the evidence base in Sierra Leone. Hand hygiene studies need to be conducted in other healthcare facilities throughout the country, and more qualitative research needs to be carried out to understand the enablers and, particularly, the barriers to better hand hygiene practice.

## 5. Conclusions

Following an operational research study in 2021 on hand hygiene compliance in two tertiary hospitals, Freetown, Sierra Leone, which led to recommendations and actions for improvement, a follow-up study on hand hygiene practices using the same methodology was conducted in 2023. Between the two years, hand hygiene compliance significantly improved in Connaught Hospital and was associated with better placement of hand hygiene reminders, good hand wash infrastructure and more frequent supervisions. Hand hygiene reminders, handwash infrastructure and supervisions were inferior at 34MH and, combined with the extensive reconstruction of the hospital at the time of the study, might have led to a general decrease in hand hygiene compliance in that hospital. This study reinforces the importance of good handwash infrastructure in promoting hand hygiene compliance. Other recommendations include the addition of innovative ways to encourage healthcare workers to wash their hands and continued quantitative and qualitative operational research to build the evidence base in healthcare facilities throughout the country and at different levels of the healthcare system. 

## Figures and Tables

**Figure 1 tropicalmed-08-00431-f001:**
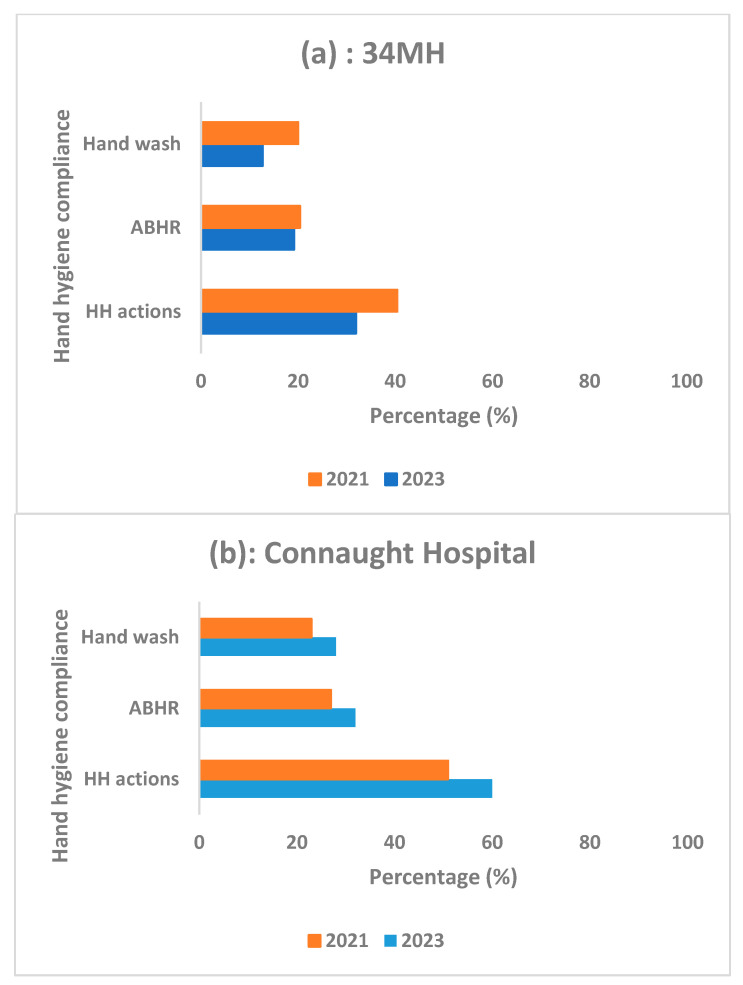
Comparison of hand hygiene compliance (and in relation to use of ABHR or hand wash) between June–August 2021 and February–April 2023 in two tertiary hospitals, Freetown, Sierra Leone: (**a**) = 34 Military Hospital (34MH); and (**b**) = Connaught Hospital. ABHR = alcohol-based hand rub; HH = hand hygiene.

**Table 1 tropicalmed-08-00431-t001:** Recommendations and actions taken to improve hand hygiene compliance between February and April 2023 in the departments and wards of the two tertiary hospitals, Freetown, Sierra Leone.

Recommendation	Action	Medicine	Surgery	Paediatrics	Accident and Emergency	Intensive Care	Obstetrics/ Gynaecology
**34MH**							
Place HH reminders/job aids in hospital departments	No. HH reminders and job aids placed on walls or tables	3	2	1	2	-	2
Improve hand wash infrastructure at hand wash stations	No. Hand wash stations	3	2	1	2	-	2
	No. Running taps	0	0	0	0	-	2
	No. Veronica buckets	3	2	1	2	-	2
	No. Receiving bowls	3	2	1	2	-	2
	No. Soap items	2	2	0	2	-	2
	No. Paper towels	0	0	0	0	-	0
	No. ABHR	1	1	0	1	-	1
Improve ABHR Supplies	ABHR supplied monthly	Yes	Yes	No	Yes	-	Yes
	ABHR missing monthly	No	No	No	No	-	No
**Connaught Hospital**							
Place HH reminders/job aids in hospital departments	No. HH reminders and job aids placed on walls or Tables	18	16	6	8	8	-
Improve hand wash infrastructure at hand wash stations	No. Hand wash stations	16	16	6	8	6	-
	No. Running taps	8	8	4	4	4	-
	No. Veronica buckets	8	8	2	4	2	-
	No. Receiving bowls	8	8	2	4	2	-
	No. Soap items	16	16	4	6	6	-
	No. Paper towels	0	0	0	0	0	-
	No. ABHR	8	8	2	2	2	-
Improve ABHR Supplies	ABHR supplied monthly	Yes	Yes	Yes	Yes	Yes	-
	ABHR missing monthly	No	No	No	No	Yes	-

Footnotes: 34MH = 34 Military Hospital; HH = hand hygiene; No. = Number; ABHR = alcohol-based hand rub.

**Table 2 tropicalmed-08-00431-t002:** Comparison of hand hygiene compliance, in relation to the five opportunities for a hand hygiene action, between June–August 2021 and February–April 2023 in two tertiary hospitals, Freetown, Sierra Leone.

Hospital and Type of HH Action	June to August 2021	February to April 2023	Change in HH Compliance 2023 v 2021	*p* Value
Opportunities for HH Action	HH Actions Performed	Opportunities for HH Action	HH Actions Performed
*n*	*n*	(%)	*n*	*n*	(%)
**34MH**								
Total opportunities:	2072	838	(40)	4529	1446	(32)	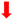	<0.001
Bef-pat	602	120	(20)	1189	140	(12)	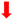	<0.001
Bef-aspet	285	89	(31)	265	25	(9)	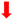	<0.001
Aft-b.f.	315	237	(75)	760	507	(68)	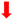	<0.001
Aft-pat	473	238	(50)	1537	593	(39)	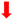	<0.001
Aft.p.surr.	397	154	(39)	778	181	(23)	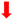	<0.001
Not recorded	-	-	-	-	-	-		
**Connaught Hospital**								
Total opportunities:	8389	4248	(51)	2677	1605	(60)	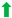	<0.001
Bef-pat	2642	672	(25)	630	151	(24)	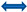	0.45
Bef-aspet	754	261	(35)	84	61	(73)	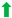	<0.001
Aft-b.f.	685	543	(79)	297	288	(97)	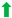	<0.001
Aft-pat	2248	1539	(69)	1119	920	(82)	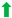	<0.001
Aft.p.surr.	2050	1228	(60)	547	185	(34)	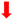	<0.001
Not recorded	10	5	(50)	0	0	(0)		

Footnotes: Observations made using the WHO hand hygiene standard observation tool [7]. HH = hand hygiene; HH action = alcohol-based hand rub or hand wash; Bef-pat = before a patient is touched; Bef-aspet = before performing a clean or aseptic procedure; Aft-b.f. = after exposure to body fluids; Aft-pat = after touching a patient; Aft.p.surr. = after touching the surroundings of a patient; 34MH = 34 Military Hospital. Percentages are row percentages. *p* values represent comparisons of hand hygiene actions between each period carried out using chi square tests.

**Table 3 tropicalmed-08-00431-t003:** Comparison of hand hygiene compliance in relation to hospital departments between June–August 2021 and February–April 2023 in two tertiary hospitals in Freetown, Sierra Leone.

Hospital and Department/Ward	June to August 2021	February to April 2023	Change in HH Compliance 2023 v 2021	*p* Value
Opportunities for HH Action	HH actions Performed	Opportunities for HH Action	HH Actions Performed
*n*	*n*	(%)	*n*	*n*	(%)
**34MH**								
Total opportunities	2072	838	(40)	4529	1446	(32)	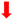	<0.001
Medical ward	401	161	(40)	911	304	(33)	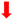	0.02
Surgical ward	555	214	(39)	960	317	(33)	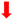	0.03
Paediatric ward	78	42	(54)	764	239	(31)	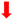	<0.001
Accident and Emergency	413	187	(45)	1118	331	(30)	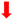	<0.001
Obstetrics/Gynaecology	468	157	(34)	776	255	(33)	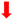	<0.001
Not recorded	157	77	(49)	0	0	(0)		
**Connaught Hospital**								
Total opportunities	8389	4248	(51)	2677	1605	(60)	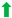	<0.001
Medical ward	3088	1427	(46)	916	562	(61)	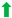	<0.001
Surgical ward	2271	1195	(53)	1002	614	(61)	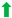	<0.001
Paediatric ward	263	165	(63)	174	114	(66)	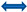	0.55
Intensive care	536	279	(52)	129	90	(70)	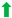	<0.001
Accident and Emergency	231	130	(56)	456	225	(49)	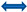	0.09
Obstetrics/Gynaecology	31	11	(36)	-	-	-		
Not recorded	1969	1041	(53)	0	0	(0)		

Footnotes: Observations made using the WHO hand hygiene standard observation tool [7]. HH = hand hygiene; HH action = alcohol-based hand rub or hand wash; 34MH = 34 Military hospital. Percentages are row percentages. *p* values represent comparisons of hand hygiene actions between each period carried out using chi square tests. 34 Military Hospital did not have an intensive care unit; Connaught Hospital did not have an obstetrics/gynaecology department in 2023.

**Table 4 tropicalmed-08-00431-t004:** Comparison of hand hygiene compliance in relation to different healthcare worker cadres between June–August 2021 and February–April 2023 in two tertiary hospitals in Freetown, Sierra Leone.

Hospital and Department/Ward	June to August 2021	February to April 2023	Change in HH Compliance 2023 v 2021	*p* Value
Opportunities for HH Action	HH actions Performed	Opportunities for HH Action	HH Actions Performed
*n*	*n*	(%)	*n*	*n*	(%)
**34MH**								
Total opportunities	2072	838	(40)	4529	1446	(32)	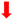	<0.001
Doctor	347	131	(38)	911	545	(60)	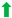	<0.001
Nurse	1582	664	(42)	2388	627	(26)	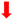	<0.001
Nursing student	62	30	(48)	564	154	(27)	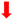	<0.001
Laboratory technician	81	13	(16)	666	120	(18)	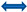	0.66
Not recorded	-	-	-	-	-	-		
**Connaught Hospital**								
Total opportunities	8389	4248	(51)	2677	1605	(60)	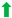	<0.001
Doctor	1892	842	(45)	633	340	(54)	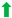	<0.001
Nurse	5112	2846	(56)	1413	856	(61)	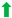	<0.001
Nursing student	1032	486	(47)	389	279	(72)	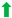	<0.001
Laboratory technician	351	72	(21)	242	130	(54)	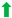	<0.001
Not recorded	2	2	(100)	0	0	(0)		

Footnotes: Observations made using the WHO hand hygiene standard observation tool [7]. HH = hand hygiene; HH action = alcohol-based hand rub or hand wash; 34MH = 34 Military Hospital. Percentages are row percentages. *p* values represent comparisons of hand hygiene actions between each period carried out using chi square tests.

## Data Availability

The datasets used in this paper have been deposited at M9.figshare.23300417 (accessed on 6 June 2023), and they are available under a CC BY 4.0 licence.

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
