# Peer review of "Have Hand Hygiene Practices in Two Tertiary Care Hospitals, Freetown, Sierra Leone, Improved in 2023 following Operational Research in 2021?"

_tropicalmed, 2023, doi:10.3390/tropicalmed8090431_

Round 1

Reviewer 1 Report (Previous Reviewer 3)

   I am satisfied with the revised version and have no other comments

Author Response

Response to Reviewer 1

Comments and Suggestions for Authors

   I am satisfied with the revised version and have no other comments

RESPONSE:

We thank the reviewer for his time and effort in helping to improve the paper and appreciate his/her response

Reviewer 2 Report (Previous Reviewer 2)

The manuscript was revised thoroughly and is now more concise and to the point. The comparison between the 2 hospitals adds to the take home message. I have no further comments.

Author Response

Response to Reviewer 2

The manuscript was revised thoroughly and is now more concise and to the point. The comparison between the 2 hospitals adds to the take home message. I have no further comments.

RESPONSE:

We thank the reviewer for his time and effort in helping to improve the paper and appreciate his/her response

Reviewer 3 Report (New Reviewer)

Matila Mattu Moiwo and colleagues present a very interesting study on a topic of the highest importance not only for Sierra Leone but also worldwide. 

The study is very well designed and the results are so valuable that they urgently need to be published.

Unfortunately, the complete presentation in the manuscript from beginning to end is still far too superficial with minor errors. This starts with the introduction, continues with tables, and continues through to the summary.

I urgently recommend major revisions to the authors. This must include the entire text, tables and figures. Furthermore, there is an urgent need to create more figures.

The authors should make these major revisions as soon as possible, as the scientific value of the paper is extremely high. 

I am already looking forward to reading the completely revised article.

Extensive language editing urgently needed 

Author Response

Response to Reviewer 3

Comments and Suggestions for Authors

Matila Mattu Moiwo and colleagues present a very interesting study on a topic of the highest importance not only for Sierra Leone but also worldwide. 

The study is very well designed and the results are so valuable that they urgently need to be published.

Unfortunately, the complete presentation in the manuscript from beginning to end is still far too superficial with minor errors. This starts with the introduction, continues with tables, and continues through to the summary.

I urgently recommend major revisions to the authors. This must include the entire text, tables and figures. Furthermore, there is an urgent need to create more figures.

The authors should make these major revisions as soon as possible, as the scientific value of the paper is extremely high. 

I am already looking forward to reading the completely revised article.

RESPONSE:

We thank the reviewer for his time and effort reviewing the paper and for his/her complements on the scientific value of the manuscript. For explanation, in a previous submission of this paper to the same journal, the Editor and the two reviewers demanded that the paper be shortened and the number of tables and figures be reduced. We duly complied with this request. In reviewing the revised version, which is what came to you, both those original reviewers were satisfied that we had addressed all their queries.

So, to answer your points. We have been thoroughly through the manuscript from start to finish and have done our best to correct any errors throughout the paper and in the tables. We really cannot add more figures or tables as we were specifically asked by the Editor and the reviewers to reduce the number in our first submission. We hope the reviewer can accept our stance here. We have made corrections in the revised manuscript using tracking changes. 

Comments on the Quality of English Language

Extensive language editing urgently needed 

RESPONSE:

Thank you. We find this comment a little difficult to address as the two other reviewers had no issues with the English and indeed the second reviewer commented that the English was fine. Nevertheless, we take this comment seriously. There is a native English speaker as co-author on the paper who is experienced in scientific paper writing and has served/ continues to serve as an Associate Editor on a number of journals. He has been carefully through the manuscript and has tried to make it as fluent and as correct as possible. 

Round 2

Reviewer 3 Report (New Reviewer)

Thanks to the implementations of the recommendations of the reviewers the manuscript has significantly improved. It is extremely important for scientists und physicians worldwide. Congratulations to the authors for this excellent research. From my point of view the article should be published as soon as possible. 

This manuscript is a resubmission of an earlier submission. The following is a list of the peer review reports and author responses from that submission.

Round 1

Reviewer 1 Report

To my opinion this manuscript does not comprise new information or highlights concerning the topic.

Reviewer 2 Report

The study is interesting and well written, yet the extent of the study and results do not justify an original article. Accordingly, the paper is too long and there is an excess of tables. I suggest narrowing of the manuscript to a brief report, and accordingly writing a more concise report. 

Reviewer 3 Report

Preforming studies regarding infection control measures in low-income countries is an important issue and I appreciate this research which was done in difficult conditions. Here are some of my comments:

1.  You can not conclude that hand hygiene reminders, infrastructure, supervisions etc. are the cause of the difference between hand hygiene rates in the two hospitals. There is association between the two phenomena but you did not show causation. Especially when the hand hygiene rates were better in Connaught Hospital in the beginning.

2. If the difference in hand hygiene rates does relate to hand hygiene reminders, infrastructure, supervisions etc. - why did it decreased in 34MH in among nurses and nursing students, but not among doctors?

3. Do you know what was the hand hygiene compliance between 9/2021 to 1/2023?

4. It is not clear whether there was an alcohol-based hand hygiene bottle on each bed. The infrastructure at hand wash stations (number of hand wash stations, running taps, veronica buckets, receiving bowls, soap items, paper towels and ABHR) should be calculated per hospital beds in order to be compared between the two hospitals

5. Is there data regarding nosocomial infections or rate of resistant bacteria during these periods of time?

6. The text is very long and sometimes repetitive, especially in the methods.